# Synthesis of Mesoporous Tetragonal ZrO_2_, TiO_2_ and Solid Solutions and Effect of Colloidal Silica on Porosity

**DOI:** 10.3390/molecules29143278

**Published:** 2024-07-11

**Authors:** Linggen Kong, Inna Karatchevtseva, Tao Wei, Jessica Veliscek-Carolan

**Affiliations:** Australian Nuclear Science and Technology Organisation, New Illawarra Road, Lucas Heights, NSW 2234, Australia; ikm@ansto.gov.au (I.K.); tao@ansto.gov.au (T.W.); jvc@ansto.gov.au (J.V.-C.)

**Keywords:** porous metal oxide, tetragonal, titanium zirconate, solid solution, hard template

## Abstract

Metal oxides possessing a large surface area, pore volume and desirable pore size provide more varieties and active industrial potentials. Nevertheless, it is very challenging to produce crystal metal oxides while keeping satisfactory porosity features, especially for ternary compositions. High temperature is usually needed to produce crystal metal oxides, which readily leads to the collapse of the pore structure. Herein, by employing a ‘soft’ dispersant agent and a hard silica template, ZrO_2_, TiO_2_ and Zr-Ti solid solutions having a tetragonal crystal structure are produced and the silica-leached materials are characterized from macroscopic to atomistic scales. The micron-sized particulate powders are composed of nanoscale ‘building blocks’, with crystallite sizes between ~8 and 21 nm. These polycrystalline ceramic powders exhibit a high specific surface area (up to ~200 m^2^·g^−1^) and pore volume (up to 0.5 cm^3^·g^−1^), with a pore size range of ~5–20 nm. Importantly, the Zr/Ti–O–Si–OH chemical bonds exist on the particle surface, with about two-thirds of the surface covered by silica. The hydroxyl groups can further post-graft organic ligands or directly associate with species. Synthesized mesoporous metal oxides are highly homogenous and could potentially be used in various applications because of their tetragonal structure and porosity features.

## 1. Introduction

Nanostructured and/or porous ZrO_2_ and TiO_2_ have been utilized in many applications due to their superior characteristics, including high mechanical, thermal and chemical stability [1,2,3,4,5,6,7]. Among various crystallographic phases, the tetragonal *t*-ZrO_2_ (space group *P*4_2_/*nmc*) and the tetragonal anatase *a*-TiO_2_ (space group *I*4_1_/*amd*) have attracted greatest interest, especially as a catalyst or catalyst support, due to their ability to attain a high specific surface area and their superior catalytic activity in comparison with either monoclinic *m*-ZrO_2_ polymorph (space group *P*2_1_/*c*) or tetragonal rutile *r*-TiO_2_ (space group *P*4_2_/*mnm*) [8,9]. In addition, it is reported that the mesoporous nanocrystalline *t*-ZrO_2_ is extremely radiation stable, with a tolerance dose up to 82 dpa without becoming amorphous [10]. As for TiO_2_, even though almost all research investigating irradiation damage has been conducted on rutile polymorph, recent studies on anatase show the crystallinity of *a*-TiO_2_ improves after neutron irradiation at normal fluence and a slight reduction in the crystallinity at the highest fluence of 4.6 × 10^17^ n·cm^−2^ (neutron energy En > 1.2 MeV), but remains higher than that of the non-irradiated material [11]. Another study shows that no phase transition towards other polymorph phases is observed even at the highest fluence studied (1 × 10^15^ n·cm^−2^) (neutron energy En > 0.1 MeV) [12]. As a result, the tetragonal crystal phase is ideally required when the ceramic nanomaterials are used as adsorbents for radioisotope generators and separation applications due to its high radiation resistance feature [13,14,15,16].

It is well understood that good material performance is usually derived from the large number of active sites, which usually are the hydroxyl groups on metal oxide surfaces. However, the preparation of *t*-ZrO_2_ and *a*-TiO_2_ with a high specific surface area (>100 m^2^·g^−1^) which are stable at elevated temperatures is challenging [8,9]. In general, high temperature treatment results in the collapse of the pore structure, which is critical for most applications. Commonly used methods to synthesize porous *t*-ZrO_2_ or *a*-TiO_2_ include sol-gel, Pechini, precipitation and hydrothermal routes [17,18,19,20,21]. One interesting stabilization mechanism of *t*-ZrO_2_ is by Si incorporation, which relates to the crystallite size effect (10–30 nm) [8]. The smearing of the siliceous species over the zirconia surface inhibits the ZrO_2_ crystallite growth and kinetically stabilizes the *t*-ZrO_2_, and the powder exhibits surface properties similar to pure silica [8,22].

In addition, it has been confirmed that one component dispersed in mixed metal oxides produces superior performance to the single metal oxide in applications such as catalysis [23,24,25] and adsorption [26]. This enhanced performance of the solid solution is due to the synergistic effect of different metal elements, and the acidity/basicity of metal oxide surfaces are varied by different mole ratios between the metals.

A direct solution combustion method [27] is reported using glucose as an organic fuel to synthesize tetragonal ZrO_2_, TiO_2_ and Zr-Ti solid solutions (Zr,Ti)O_2_. A full crystallization of primarily single phase with the tetragonal structure is produced at 600 °C calcination for 4 h for ZrO_2_, TiO_2_ and their mixture with TiO_2_ ≤ 33 mol% and crystallite size < 30 nm. However, the specific surface areas of the obtained powders are <50 m^2^·g^−1^, which is relatively low for real applications. A ternary metal oxide (Y_2_Ti_2_O_7_) possessing a pyrochlore crystal structure is synthesized using a novel technique which combines ‘soft’ and hard colloid templating strategies [28]. The materials are composed of ~25–30 nm ‘building blocks’ with high chemical and phase stability. The polycrystalline powders have a specific surface area up to 70 m^2^·g^−1^ and pore volume of ~0.35 cm^3^·g^−1^, after 900 °C calcination for 6 h. Using the same synthesis method, a cerium titanate CeTi_2_O_6_ having a brannerite crystal structure is produced [29]. The synthesized powders exhibit similar particle morphologies and porosity features as those of the porous pyrochlore [28].

This study aims at a synthesis of hierarchical mesoporous zirconia, titania and Zr-Ti solid solutions possessing a tetragonal crystal structure. The glucose is used as a fuel/dispersant agent and colloidal silica is introduced as a hard template. High porosity of the metal oxides remains after removing the ‘soft’ and hard templates. Crystal structures and morphologies of the materials are analyzed using X-ray diffraction, spectroscopy, electron microscopy and nitrogen porosimetry. The primary goal of this work is to achieve a high specific surface area (>100 m^2^·g^−1^) for synthesized powders to pave the way for potential applications of materials requiring high radiation resistance.

## 2. Results and Discussion

### 2.1. XRD Analysis

A previous study [27] shows that calcination at 700 °C leads to the formation of the minor phases *m*-ZrO_2_ and *r*-TiO_2_ for pure ZrO_2_ and TiO_2_, respectively. However, for Zr-Ti solid solutions, the crystal structure is hardly changed, remaining in a pure tetragonal phase at 700 °C. In addition, an increase to the Ti proportion (≥50 mol%) in Zr-Ti solid solutions results in an incomplete crystallization, which is evident in very broad XRD peaks. As a result, in this study, calcination at 600 °C is selected to produce *t*-ZrO_2_, *a*-TiO_2_ and *t*-Zr-Ti solid solutions with Ti ≤ 25 mol%.

Since the purposes of this work are to achieve a higher specific surface area and to focus on the Zr-Ti solid solutions while the powders possess a tetragonal structure, two series of samples are investigated, as shown in Table 1. One set of samples is composed of ZrO_2_, Zr_0.875_Ti_0.125_O_2_ (Z_7_T_1_), Zr_0.75_Ti_0.25_O_2_ (Z_3_T_1_) and TiO_2_, with a silica-to-(Zr,Ti)O_2_ weight ratio of 1:2. Another series of samples are Z_3_T_1_ solid solutions with silica-to-Z_3_T_1_ weight ratios ranging from 0 to 1:1. Figure 1 shows the XRD patterns of the silica-leached powders, with all samples possessing a tetragonal crystal structure. Both ZrO_2_ (Z-1/2) and Zr-rich (Zr,Ti)O_2_ (Z_7_T_1_-1/2 and Z_3_T_1_-1/2) have a tetragonal structure with XRD reflections at (101), (002), (110), (102), (112), (200), (103), (211), (202) and (220) in the range of 25–80° (2*θ*). The crystallite size of ZrO_2_ is calculated using the Scherrer equation to be 7.9 nm, and it increases to 8.9 and 10 nm (Table 1) when Ti fractions increase to 12.5 and 25 mol%, respectively. A larger crystallite size is generally desired, so Zr_3_Ti_1_ composition materials with varying silica content are used for further study. TiO_2_ (T-1/2) represents a pure tetragonal anatase phase with XRD reflection peaks indexed as (101), (004), (200), (105), (211), (204), (116), (220) and (215), and the crystallite size is 7.7 nm. The crystallite sizes of the counterpart powders having identical chemical compositions without the introduction of silica are in the range ~23–25 nm [27], suggesting the presence of silica inhibits crystal growth.

The effect of silica on the crystallization of the solid solution Z_3_T_1_ is further investigated. The powder without addition of the silica (Z_3_T_1_-0) exhibits sharp and intense XRD peaks, indicating higher crystallinity of the material and a larger crystallite size (21 nm). With a gradual increase in silica-to-Z_3_T_1_ weight ratios from 1:4 (Z_3_T_1_-1/4) to 1:3 (Z_3_T_1_-1/3) and 1:2 (Z_3_T_1_-1/2), all powders indicate the tetragonal structure; however, the broadening of XRD peaks at higher weight ratios suggests structural distortion and/or smaller crystallite sizes. It is apparent that the crystallite size decreases progressively from 21 nm for Z_3_T_1_-0 to 12, 11 and 10 nm when silica-to-Z_3_T_1_ weight ratios increase from 1:4 to 1:3 and 1:2. When the weight ratio reaches 2:3, the XRD pattern shows a low degree of crystallinity, with only a few XRD peaks observed, and the crystallite size is calculated to be 4.8 nm. Further increasing the silica such that it equals Z_3_T_1_ in weight leads to a very low crystallized powder, but not a completely amorphous phase, as the crystallite size is estimated, with low confidence, to be 1.4 nm and the very broad XRD peaks indicate the loss of the long-range order. The formation of the stable tetragonal phase is important for nuclear applications, which generally require the applied materials to have high radiation resistance features.

These results with respect to Z_3_T_1_ materials are similar to phenomena observed by Abel et al., who investigated the silica effect on the formation of *t*-ZrO_2_ calcined at 650 °C with different Si mole fractions (0–27 mol%) [8]. In their study, Ludox^®^ AS 40 colloidal silica is used, which has a surface area of ~135 m^2^·g^−1^ and a calculated colloid size of ~20.3 nm. Without Si, *m*-ZrO_2_ is predominantly formed. Almost pure *t*-ZrO_2_ is obtained at 8 mol% Si and above. At higher Si molar fractions, *t*-ZrO_2_ reflections become visibly broad, until an X-ray amorphous material is formed at 27 mol% Si. It is also observed that as the Si mole fraction increases, the calculated crystallite sizes decrease from 23.7 nm (pure ZrO_2_) to 7.5 nm (19 mol% Si). As a result, they conclude that no XRD reflections observed at 27 mol% Si mole fraction are likely due to even smaller crystallite sizes, as an amorphous, growth-constraining SiO_2_ network forms around the ZrO_2_ crystallites [8]. In our case, for Z_3_T_1_ samples, distinct tetragonal structures are formed until a Si mole fraction at 48.3 mol% (sample Z_3_T_1_-1/2 in Table 1), weaker and broader XRD peaks are observed for Z_3_T_1_-2/3 with 55.5 mol% Si and the Z_3_T_1_-1/1 powder becomes the least crystallized at 65.2% Si mole fraction, possibly due to a very small crystallite size. Hence, using colloidal silica of different sizes and/or incorporating Ti in zirconia enable the tetragonal crystal structure to be maintained at higher Si mol fractions than in the previous investigation of Abel et al. [8]. Appendix A shows the XRD patterns of the calcined powders prior to leaching silica hard templates. These patterns are similar to those of the counterpart samples shown in Figure 1, suggesting silica is in an amorphous state and does not incorporate inside the tetragonal crystal structure.

The crystallite size of *t*-(Zr,Ti)O_2_ formed on a colloidal silica particle surface depends on the concentration of (Zr,Ti)O_2_ per nm^2^ of silica surface. A high number (*N*) of (Zr,Ti)O_2_ molecules on the silica surface (*N*·nm^−2^) would lead to a bigger crystallite size. Taking Z_3_T_1_ composition as an example, there are 97 Z_3_T_1_ molecules per nm^2^ of silica surface for Z_3_T_1_-1/4. When the number of molecules of Z_3_T_1_ is ~37 (Z_3_T_1_-2/3), the crystallite size is only ~4.8 nm. Very small crystallites will be formed when the number of molecules is lower than ~30 per nm^2^ of silica surface. The formation mechanisms of *t*-ZrO_2_ or *t*-Z_3_T_1_ in the presence of Si may be different with respect to Si introduced as colloidal silica (SiO_2_) and silicon alkoxide at molecules level. When silica nanoparticles are used, the interaction between Si and Zr or Zr-Ti occurs on the surface of the silica colloids. The silanol groups (Si–O–H) on the silica surface are functional active sites for any chemical reactions. The hydroxyl groups (–OH) on colloidal silica react with hydroxyl groups on zirconia (Zr–OH) followed by a dehydration condensation reaction, resulting in the formation of chemical bonds between ZrO_2_ and silica (Zr–O–Si). In this case, the greater the number of silica colloids presented, the higher the surface area of the silica and, as such, the greater the number of reaction sites. Under a fixed amount of the Zr or Zr-Ti molecule, fewer Zr or Zr-Ti reactants presented in a fixed silica surface area lead to smaller crystallite or crystal grains.

Previous studies show that the introduction of silicon is able to stabilize *t*-ZrO_2_ [30,31,32,33]. The stabilization is attributed to either the constraining effect of the amorphous silica on ZrO_2_ particles at high SiO_2_ content (>30% mol) [30,31] or the decreased kinetics of ZrO_2_ grain growth induced by silica as low as 2 mol% SiO_2_ [32,33]. High-temperature calcination normally leads to the formation of *m*-ZrO_2_ [27]. When a small amount of silica is introduced, it does not crystallize but forms silica-rich glassy phase concentrates at the triple junctions of the *m*-ZrO_2_ crystallites [34]. This amorphous silica-rich phase contains many substitution defects and might have considerable acidity [35]. So the introduction of silica will increase the phase transformation temperature from tetragonal to monoclinic.

Because the silicon atom is small and strongly prefers tetrahedral coordination, its dissolution within the ZrO_2_ lattice seems less probable than at the superficial location [22]. It is evidenced [22] that the zirconia textural can be stabilized by siliceous species, and this effect is due to smearing of the siliceous species over the precipitate surface. This support surface is strongly enriched by silicon; at a Si content as low as 4 wt%, the solids show properties similar to pure silica catalytic supports for both noble metal- and sulfide-catalyzed reactions [22], indicating Si-stabilized zirconia possessing a substantial surface area and thus a high concentration of active sites (–OH) [8]. This observation is very important for further reaction with other species via either hydrogen bonding or electrostatic force.

### 2.2. Raman and FTIR Analyses

Complementary to XRD analysis, which focuses on long-range cation order–disorder of the crystal structure, the Raman technique is used to study the short-range atomic arrangement of the metal oxides by monitoring the metal–oxygen vibrations. The active modes for *t*-ZrO_2_ polymorph with space group *P*4_2_/*nmc* ≡ D*_4h_* include six Raman peaks (*A*_1*g*_ + 2*B*_1*g*_ + 3*E_g_*) and three IR bands (*A*_2*u*_ + 2*E_u_*), with two molecules per unit cell [36]. Raman spectra are generally well defined and very useful for characterization of the metal oxides, while middle range IR spectra are more difficult to interpret.

Figure 2 shows the Raman spectra of two series of powders after leaching silica. The zirconia (Z-1/2) displays well-defined Raman peaks, including 148, 270, 314, 461 and 642 cm^−1^, and weak shoulder peak at ~604 cm^−1^. These peak positions of the silica-leached powder are very similar to those of the pure ZrO_2_ sample (146, 272, 316, 461, 645 and shoulder 604 cm^−1^) studied previously [27], suggesting that the presence of silica during formation of ZrO_2_ has no apparent effect on the short-range order; i.e., Zr–O vibrations. More importantly, a weak and broad peak detected at ~190 cm^−1^ in the pure ZrO_2_ sample is highly likely due to *m*-ZrO_2_ phase [27]; however, it is barely visible for silica-leached ZrO_2_ (Z-1/2). This may indirectly confirm that silica might increase the temperature of phase transformation from *t*- to *m*-ZrO_2_. The *E_g_* mode at 148 cm^−1^ is assigned to O–Zr–O and Zr–O–Zr bending vibrations. The 270 cm^−1^ band is the *A*_1*g*_ mode, which is mainly due to the Zr–O stretching vibration. Raman bands at 461 cm^−1^ (*E_g_* mode) and 314 cm^−1^ (*B*_1*g*_ mode) can be assigned to a combination of bending and stretching vibrations. The 642 cm^−1^ peak (*E_g_* mode) and very weak shoulder at 604 cm^−1^ (*B*_1*g*_ mode) are attributed to the Zr–O stretching vibrations [27].

Tetragonal *a*-TiO_2_ is also characterized by six Raman active vibrations (*A*_1*g*_ + 2*B*_1*g*_ + 3*E_g_*) [27]. Sample T-1/2 exhibits three *E_g_* modes located at 145, 199 and 638 cm^−1^. The *B*_1*g*_ mode is found at 397 cm^−1^ and a combination band (*A*_1*g*_ + *B*_1*g*_) is observed at 517 cm^−1^. These Raman peak positions are similar to those observed for pure titania powder (144, 195, 397, 517 and 640 cm^−1^) [27], indicating that silica does not affect the formation of the *a*-TiO_2_ crystal structure.

When 12.5 and 25 mol% of Zr are replaced by Ti for Z_7_T_1_-1/2 and Z_3_T_1_-1/2, respectively, the location of two *E_g_* modes at ~640 and 148 cm^−1^ is hardly changed; however, peak broadening is clearly noticeable for both Raman bands. The major Raman band centered at ~320 cm^−1^ appears in both samples and most likely is due to a combination of Zr–O and Ti–O stretching vibrations. The broadening of this Raman peak may be due to the deformation of the tetragonal structure with the incorporation of Ti in a ZrO_2_ unit cell, rather than the crystallite size effect. The Raman spectra of the silica-leached samples with various silica-to-Z_3_T_1_ weight ratios are also shown in Figure 2. Both crystallite size and structure deformation have impacts on the Raman spectra for these powders. The continuous broadening of the major peak observed at ~320 cm^−1^ is due to the gradual decrease in the crystallite size [37]. With a further increasing of the silica-to-Z_3_T_1_ weight ratio to ≥2:3, the Raman spectrum becomes featureless. The extremely small crystallite size could be responsible for this observation, in addition to the dis-ordered nature of these samples because of the incorporation of 25 mol% of Ti.

The IR spectroscopy is employed to determine if Zr/Ti–O–Si bonds are formed at the Zr/Ti oxide interface. The principal absorption frequencies in both ZrO_2_ and TiO_2_ are routinely observed in the IR region below 800 cm^−1^. The IR spectrum of SiO_2_, on the other hand, is characterized by several main absorption bands, including a very strong peak at 1100–1000 cm^−1^ owing to the Si–O–Si asymmetric stretching vibration [38,39]. The FTIR spectra of the silica-leached powders are presented in Figure 3 and Appendix A. As expected, all samples display a strong IR absorption below 850 cm^−1^, which is due to the M–O–M (M = Zr or Ti) bond vibration modes.

From Figure 3, the IR spectra display strong absorption at 957–941 cm^−1^, reflecting the presence of the O–Si bond. It is noted that the frequency of the O–Si stretching band in these synthesized materials is significantly lower than the band corresponding to the same type of vibration in amorphous silica (~1100 cm^−1^) [38,39]. This behavior manifests the presence of an elevated number of Zr–O–Si and/or Ti–O–Si bonds in the samples. The phenomena have been observed for ZrO_2_-SiO_2_ powders [33,40,41]. The IR peak ~990 cm^−1^ is observed for amorphous ZrO_2_-SiO_2_ powders prepared by hydrolysis of a liquid aerosol and dried at 50 °C [40,41]. For ZrO_2_/SiO_2_ binary oxides thermally treated at 600 °C [33], the typical stretching peak of Zr–O–Si shifts progressively from 1075 cm^−1^ for silica-rich (10 mol% ZrO_2_ + 90 mol% SiO_2_) to 1002 cm^−1^ for zirconia-rich (90 mol% ZrO_2_ + 10 mol% SiO_2_) materials. In addition, it is reported that the band corresponding to Ti–O–Si vibrations is usually observed in the 950–970 cm^−1^ zone for TiO_2_–SiO_2_ nanocomposites [42], which is also lower than 1100 cm^−1^.

With the gradual increase of the Ti component in zirconia samples Z-1/2, Z_7_T_1_-1/2 and Z_3_T_1_-1/2, the IR band responsible for O–Si bond vibrations not only shifts to lower wavenumbers, but also becomes less intense. These results suggest that the Zr/Ti ratio has a strong impact on O–Si bond vibrations when it is chemically bonded with Zr/Ti. ZrO_2_ and TiO_2_ have a similar specific surface area in our study, which will be discussed later. The intense IR peak at ~957 cm^−1^ for Z-1/2 may indicate that zirconia has a stronger affiliation with silica than titania (T-1/2).

The weak-to-medium IR peaks at 1348–1360 cm^−1^ (Figure 3) are ascribed to δ(Si–OH) groups of hydrated silica. This band is routinely observed at ~1400 cm^−1^ [39] and its shifting to lower wavenumbers may indicate that Si–OH groups are chemically bonded on (Zr,Ti)O_2_ surfaces. Furthermore, weak and broad IR bands located at ~1600 cm^−1^ are due to free –OH vibrations (water within the metal oxide structure) [39].

It is expected that powder materials with a higher specific surface area will account for a higher number of Zr/Ti–O–Si and Si–OH species. Thus, for Z_3_T_1_ solid solutions with a higher amount of silica, intense IR peaks at 941 and 1348 cm^−1^ suggest higher concentrations of surface Si–O–Si and Si–OH groups. For material without the addition of silica, both IR peaks are not observed.

When these porous materials are used as adsorbents for nuclear applications such as radionuclide generation and radioactive waste separation, the presence of the active Si–OH groups on particle surfaces is of great importance. The high number of –OH groups can directly associate with radioactive species with high efficiency; they can also further graft organic ligands to improve selectivity.

### 2.3. SEM and TEM Analyses

The overall morphology of the powders imaged by secondary SEM is presented in Figure 4. The particulate shape is irregular and the size ranges from sub-micron to tens of microns. The high magnification SEM images (Figure 4 inset) display an interconnected porous network structure (like ‘necked millet seeds’) with the mesopores originated from remaining voids after leaching silica. The glucose introduced acts as a dispersant agent during drying and calculation, leading to the formation of the aggregated/agglomerated crystalized particles, not a completely gelled monolith. The hydroxyl groups on the glucose may associate with the metal oxide surfaces by a weak chemical bond. On the other hand, the nonpolar organic part of the glucose on the fringe of the complex-building species yields steric repulsion, which inhibits the complete coalescence/’fusion’ of the particulates [27,28,29]. It is expected that particulate morphology, including particle size and size range, can be controlled by judiciously selecting the type and quantity of the dispersant agent.

The energy dispersive spectroscopy (EDS) results analyzed by SEM for samples with different compositions are given in Table 2. Element Na is derived from the remaining NaOH used during the leaching process. A high level of Na (5.1 mol%) for Z_3_T_1_-1/1 is possibly due to the high surface area of the sample. A thin layer of SiO_2_ is chemically bonded on Zr/Ti oxide surfaces via a Zr/Ti–O–Si bond, as evidenced by IR results (Figure 3). SiO_2_ is more likely to associate with the ZrO_2_ surface (2.7 mol% Si) than with TiO_2_ (0.7 mol% Si), which is also consistent with the IR observation. It is not clear whether the bonding between silica and the (Zr,Ti)O_2_ surface is homogeneous or not, but the smearing of the siliceous species leads to the powder exhibiting properties similar to pure amorphous silica [8,22], which is then easy to be functionalized for industrial applications. In addition, it is not surprising that more Si on sample Z_3_T_1_-1/1 (5.1 mol% Si) is observed than on Z_3_T_1_-1/2 (2.5 mol% Si), as the former possesses a higher specific surface area.

The microstructures of the samples after removing silica and the primary grain sizes of the particulates are observed using TEM. Figure 5 and Appendix A show the TEM bright field images of the particles at different magnifications. The TEM images display ceramic particulates composed of the aggregated primary nanoparticles with diameters of ~5–10 nm (Appendix A). This observation is consistent with the crystallite estimation obtained by XRD analysis.

Figure 6 shows the selected area diffraction rings generated from the selected aperture area, which includes many nano-sized grains with different crystal orientations. The patterns belong to the tetragonal crystalline structure. The Z-1/2 and Z_3_T_1_-1/2 exhibit the same SAED ring patterns; i.e., (101), (002), (112) and (211) with space group *P*4_2_/*nmc*. Z_3_T_1_-1/1 only shows the three inner rings, indicating the same crystal structure as Z-1/2 and Z_3_T_1_-1/2, but relatively lower crystallinity and/or smaller crystallites. T-1/2 shows the ring pattern with four diffraction rings; i.e., (101), (004), (200) and (121) in space group *I*4_1_/*amd*. The space distances calculated from the (101) plane ring via TEM are 0.297, 0.296 and 0.354 nm for Z-1/2, Z_3_T_1_-1/1 and T-1/2, respectively, whereas the d-spacings of the (101) plane measured by XRD are 0.2929 nm (2*θ* = 30.523°), 0.2938 nm (2*θ* = 30.424°) and 0.3483 nm (2*θ* = 25.573°) for the corresponding samples. The d-spacing of the (002) plane for Z_3_T_1_-1/2 is estimated by TEM to be 0.261 nm while the value is measured by XRD to be 0.2599 nm (2*θ* = 34.516°).

HRTEM images shown in Figure 7 demonstrate the lattice structures of the crystal. The d-spacings of lattice planes based on the HRTEM estimation are in good consistency with XRD data, with a value discrepancy of <2%. The results confirm the full crystallization of the materials for most samples, with crystallite grains exhibiting a similar scale to the primary particles. Although the TEM images and SAED patterns show the specimen features of small areas whereas the XRD results reflect the characteristics of the bulk materials, the d-spacing measurements by TEM match well with the XRD results. As a result, both techniques are used complementarily to better understand the microstructure and overall structure of the materials.

### 2.4. Nitrogen Sorption Analysis

Specific surface area, pore volume and pore size of the silica-leached powders are determined by nitrogen sorption analysis. The adsorption and desorption isotherms, the corresponding pore width distribution and cumulative pore volume with respect to pore diameter are shown in Figure 8 for tetragonal ZrO_2_, TiO_2_ and two solid solutions, with a 1:2 silica-to-(Zr,Ti)O_2_ weight ratio. All sorption diagrams exhibit the characteristic features of the Type IV isotherm using the IUPAC classification scheme [43,44], and the hysteresis loops correspond mainly to the presence of the mesopores (2–50 nm). The detailed porosity data are summarized in Table 3. In general, there is no significant variation for specific surface area, which is in the range of 171–191 m^2^·g^−1^ and 166–187 m^2^·g^−1^ as analyzed by BET and DFT, respectively. The pore volume for ZrO_2_ (Z-1/2) is ~0.3 cm^3^·g^−1^, which increases by ~10% when 12.5 mol% of Zr is replaced by Ti (Z_7_T_1_-1/2). Further increasing the Ti content in a solid solution (Z_3_T_1_-1/2) leads to a higher pore volume, ~0.36 cm^3^·g^−1^, a similar value to pure TiO_2_ (T-1/2). The pore size distribution profiles shown in Figure 8 demonstrate predominantly mesoporous pores (2–50 nm), with the volume being ~97–99% of the total pores (Table 3). The average pore size estimated by BET is ~7–9 nm, while the pore size determined by DFT is ~10 nm for most samples. The contribution from micropores (pore size < 2 nm) to pore volume is less than 2.2%. In most cases, there are two pore size peaks at ~5 and 10 nm within the mesopore size range. It is expected that the 5 nm pores are derived from the voids between (Zr,Ti)O_2_ crystallites, while 10 nm pores are originated from the spaces occupied by colloidal silica before leaching. Overall, the Zr/Ti ratio has minimal impact on the porous structure under a fixed silica-to-(Zr,Ti)O_2_ weight ratio.

For silica-leached Z_3_T_1_ samples at various silica-to-Z_3_T_1_ weight ratios, the nitrogen sorption isotherms, pore size distributions and corresponding cumulative pore volumes are shown in Appendix A and detailed pore structure data are listed in Table 4. All isotherm diagrams demonstrate Type IV nitrogen sorption characteristics [43,44]. By increasing the silica-to-Z_3_T_1_ weight ratio from 0 to 1:1, the specific surface area increases from approximately 20 to 230 m^2^·g^−1^. Specifically, a linear relationship is expressed between the specific surface area and a silica/Z_3_T_1_ weight ratio in the range of 0 to 0.5 (Figure 9a). With a further increase to the weight ratio of 0.5 to 1.0, the specific surface area still increases gradually, but at a slower pace compared to samples having less silica.

In addition, it is apparent that total pore volume is increased linearly with respect to the silica-to-Z_3_T_1_ weight ratio within the 0–0.67 range (Figure 9b). The pore volume of the sample without adding silica is 0.04 cm^3^·g^−1^. When the silica-to-Z_3_T_1_ weight ratio is 2:3 (Z_3_T_1_-2/3), the pore volume increases to ~0.45 cm^3^·g^−1^, which is about 11 times higher than that of Z_3_T_1_-0. Further increasing the weight ratio from 2:3 to 1:1, the pore volume increases by ~10% to ~0.50 cm^3^·g^−1^. The average pore diameter estimated by BET is ~8 nm, whereas the pore size determined by DFT is about 10 nm for all silica-leached samples,

For Z_3_T_1_-0, there is only one pore size peak of ~7 nm (Appendix A), which is the void spaces between crystalline grains. For all silica-leached samples, two pore size peaks are observed at ~5 and 10 nm (Appendix A). The smaller pores are the voids between ceramic grains, while the larger pores are the remaining spaces after leaching silica.

It can be seen from SEM and TEM images (Figure 4 and Figure 5) that the particulates of the powder are composed of nanocrystalline ‘building blocks’/primary particles. The specific surface area of the porous powder determined by nitrogen sorption is mainly contributed by the surface area of the nano grains, even though there might be open pores on the grains which would be in small proportion. The total surface area of the crystallites can be calculated, with the results shown in Appendix A, under the assumption that the crystals are spherical. For Z_3_T_1_-1/4, Z_3_T_1_-1/3 and Z_3_T_1_-2/3, the calculated figures are close to the experimental results, suggesting that the crystal grain size is close to the crystallite size. The calculated figure for Z_3_T_1_-0 is more than double the experimental value, indicating that each crystal grain consists of several crystallites, which can be observed from the TEM images. For Z_3_T_1_-1/2, the higher value of the experimental result compared to the calculation may suggest that the real grain size is smaller than the calculated crystallite size and/or that crystal grains are highly porous. For Z_3_T_1_-1/1, the estimated crystallite size is ~1.4 nm, which may be much smaller than the true value. As a result, a higher calculated surface area is obtained.

Certainly, one (Zr,Ti)O_2_ grain may contain several crystallites and colloidal silica may undergo aggregation and/or coalescence during sample preparation. In both cases, the surface areas of the (Zr,Ti)O_2_ grains and the number of the silica nanoparticles will be decreased. In general, the silica covers ~60–70% of the surface area of Z_3_T_1_ powder, regardless of the silica-to-Z_3_T_1_ weight ratios (except low-confidence sample Z_3_T_1_-1/1). Hence, it is reasonable to assume that the presence of silica inhibits the growth of the ceramic grains and that approximately two-thirds of the Z_3_T_1_ surface area is covered by silica (Appendix A).

The tetragonal ZrO_2_, TiO_2_ and Zr-Ti solid solutions are synthesized using a solution combustion method in the presence of an organic fuel only [27]. The maximum specific surface area of ~50 m^2^·g^−1^ is observed for ZrO_2_, which decreased to ~34 and ~10 m^2^·g^−1^ for Zr_0.75_Ti_0.25_O_2_ and TiO_2_, respectively. The pore volumes are <0.04 cm^3^·g^−1^ for all these powders. By combining a ‘soft’ dispersant and a hard colloidal template, crystalline ternary metal oxides have been produced. Pyrochlore Y_2_Ti_2_O_7_ powders [28] are composed of ~25–30 nm nanoparticles, with a specific surface area up to 70 m^2^·g^−1^ and a pore volume of ~0.35 cm^3^·g^−1^. Brannerite CeTi_2_O_6_ polycrystalline materials [29] are assembled by primary particles of ~20–30 nm and exhibit a slightly higher specific surface area (up to ~100 m^2^·g^−1^) and pore volume (~0.4 cm^3^·g^−1^) than those of the porous pyrochlore [28]. In the present work, the pore structure features have been significantly improved compared with previous studies [27,28,29]. The optimal sample in terms of porosity and crystallinity was Zr_3_Ti_1_-1/2, as this sample demonstrates the highest surface area and pore volume while still maintaining a tetragonal crystal structure. Increasing the silica content further increases the porosity but results in poor crystallinity. More importantly, the existence of the Zr/Ti–O–Si bonds and, thus, the active Si–OH groups on the surface of the materials pave the way for further studies. As a result, these materials attain several critical features for potential applications. A tetragonal crystal structure is radiation resistant, and its high porosity ensures a high concentration of –OH groups on the particle surface, which provides active sites for post-grafting organic ligands or directly conjugating with other species.

## 3. Materials and Methods

### 3.1. Materials

Zirconyl nitrate hydrate [ZrO(NO_3_)_2_∙*x*H_2_O] (99 + %), titanium(IV) (triethanolaminato) isopropoxide [Tyzor TE] (compound dissolved in 2-propanol) and d-(+)-glucose (99.5 + %), colloid silica aqueous suspension [Ludox^®^ HS-30; surface area ~220 m^2^·g^−1^, calculated colloid diameter Φ ~12.4 nm using density as 2.196 g·cm^−3^] were purchased from Sigma-Aldrich (Saint Louis, MO, USA) and used as received. The Zr and Ti contents of the raw materials were quantitatively determined by gravimetric method and an Agilent 7900 inductively (Agilent Technologies, Santa Clara, CA, USA) coupled plasma mass spectrometer (ICP-MS), respectively. Milli-Q grade water was used in all experimental procedures.

### 3.2. Methods

Totals of 20 mmol of stoichiometrically calculated zirconyl nitrate and Tyzor TE were dissolved in 50 mL water with added glucose (glucose: metal oxide = 1:1 *w*/*w*). A clear solution was produced upon stirring for 30 min at 45 °C. Calculated Ludox^®^ HS-30 aqueous suspension was added and subsequently stirred for 1 h to form a homogeneous mixture, which was dried overnight in an oven at 100 °C. The dried gelling material was calcined in a furnace in air at 600 °C for 6 h with a 2 °C min^−1^ ramp rate and a 5 °C min^−1^ cooling rate. The calcined powders were stirred in 2.5 mol L^−1^ NaOH for >8 h at 40 °C (30 mL of basic solution per 1 g powder) in a polypropylene bottle. This leaching process was repeated three times to remove the silica template. The resultant particulates were washed twice with water, followed by centrifugation and drying overnight in an oven at 100 °C. The sample name and the corresponding composition, silica-to-(Zr,Ti)O_2_ weight ratio and mole ratio, and silica mole fraction are listed in Table 1.

### 3.3. Characterization

XRD patterns were collected on a PANalytical X’Pert Pro diffractometer (Almelo, The Netherlands) using Cu K_α_ radiation (*λ*_av_ = 1.54187 Å) at 45 kV and 40 mA. The data were recorded over an angular range of 10–80° (2*θ*) with a step size of 0.03° and an acquisition time of 2 s per step. The average crystallite size (*L*) was estimated from the line broadening of the most intense XRD peak (101) using the Scherrer formula.
$$L=\frac{0.9\lambda }{\beta \cdot \cos\theta }$$
where *λ* refers to the wavelength of the X-rays, *β* the full width at half-maximum height of the (101) peak (rad), *θ* the diffraction angle and 0.9 the shape factor.

Raman spectra were used to investigate the local structure of the materials using a Renishaw inVia Raman spectrometer equipped with an Argon ion laser (532 nm) and a Peltier cooled CCD detector (Renishaw plc, Old Town, Gloucestershire, UK). Stokes-shifted Raman spectra were collected in the static mode in the range of 100–700 cm^−1^, with a spectral resolution of 1.7 cm^−1^ for the 1800 L/mm grating. On average, 20 spectra were collected for each sample. The spot size was approximately 1.5 μm for 50× magnification.

Fourier transform infrared spectroscopy (FTIR) spectra were obtained on a Nicolet iS50 FTIR spectrometer (Thermo Electron Corporation, Madison, WI, USA) equipped with a Polaris™ mid-IR source and DTGS ATR detector. Mid-IR spectra were collected using a built-in, all-reflective diamond ATR (Attenuated Total Reflectance) sampling accessory. Spectra were collected from 400 to 4000 cm^−1^ at 4 cm^−1^ resolution with 16 scans acquired for each sample.

Scanning electron microscopy (SEM) and energy dispersive spectroscopy (EDS) were used to analyze the particle morphology and phase compositions. The powder samples were examined by a Zeiss Ultra SEM (Carl Zeiss NTS GmbH, Oberkochen, Germany) operating at 15 kV and equipped with an Oxford Instruments X-Max 80 mm^2^ SDD X-ray microanalysis system. Samples were coated with carbon (~5 nm) prior to examination. EDS analyses were performed on Oxford Instruments INCA software (V5.05). A copper metal standard was used for Quant Optimization to ensure the best fit of the stored profiles within the software.

Transmission electron microscope (TEM), selected-area electron diffraction (SAED) pattern and high-resolution transmission electron microscope (HRTEM) analyses were performed to obtain more detailed information about the crystal structure of the powder particles. Samples for TEM analyses were prepared by suspending powder in ethanol, followed by deposition onto carbon-coated 200-mesh copper grids. A JEM 2200FS (JEOL Ltd., Akishima, Tokyo, Japan) TEM device operating at 200 kV was used to conduct TEM, SAED and HRTEM examinations.

Surface area, pore volume and pore size were determined using nitrogen adsorption analysis at 77K on a Autosorb IQ volumetric adsorption analyzer (Quantachrome Instruments, Boynton Beach, FL, USA). Samples were outgassed overnight at 150 °C to remove CO_2_ and water. The N_2_ adsorption–desorption isotherms were interpreted by the IUPAC classification scheme [43,44]. Specific surface area and pore volume are modeled by both conventional Brunauer–Emmett–Teller (BET) theory and molecular simulation density functional theory (DFT). Although, from a scientific point of view, the assumptions made in BET theory do not consider micropore filling, it is still widely accepted by the research community for comparison purposes. DFT, on the other hand, considers the mechanisms of micropore filling as well as of pore condensation, evaporation and hysteresis in mesopores, and can be employed to calculate a reliable pore size distribution over the complete micropore and mesopore range.

## 4. Conclusions

Zirconia, titania and selected Zr-Ti solid solutions are synthesized via soft chemistry route using glucose as a dispersant agent and colloidal silica as a hard template. The synthesized powders possess tetragonal crystal structure and hierarchical mesoporous features. The effect of the silica-to-Zr_0.75_Ti_0.25_O_2_ weight ratios on crystallinity and porosity is investigated. XRD, Raman and TEM confirm that a full crystallization microstructure with the single tetragonal phase is obtained for silica-leached powders after calcination at 600 °C. The crystallite size is in the range of ~8–21 nm, depending on the compositions and the silica-to-(Zr,Ti)O_2_ weight ratios. The formed materials range from sub-micron to tens of microns in size and are composed of ‘building blocks’ in nanometers. FTIR and EDS-SEM results indicate that a small percentage of silica remains in the final product, depending on the surface area. The nitrogen sorption results show that the template-leached polycrystalline powders have a high specific surface area ranging from ~100 to 200 m^2^·g^−1^, and the corresponding pore volumes are ~0.2–0.5 cm^3^·g^−1^. The powders are mesoporous, with most pores in the size range of ~5–20 nm. The presence of silica inhibits the growth of the (Zr,Ti)O_2_ crystal grains, and about two-thirds of the Zr_0.75_Ti_0.25_O_2_ surface area is calculated to be covered by silica. These mesoporous powders could be used for nuclear applications such as generator adsorption and separation, due to the radiation tolerance of the tetragonal crystal structure and the high specific surface area.

## Figures and Tables

**Figure 1 molecules-29-03278-f001:**
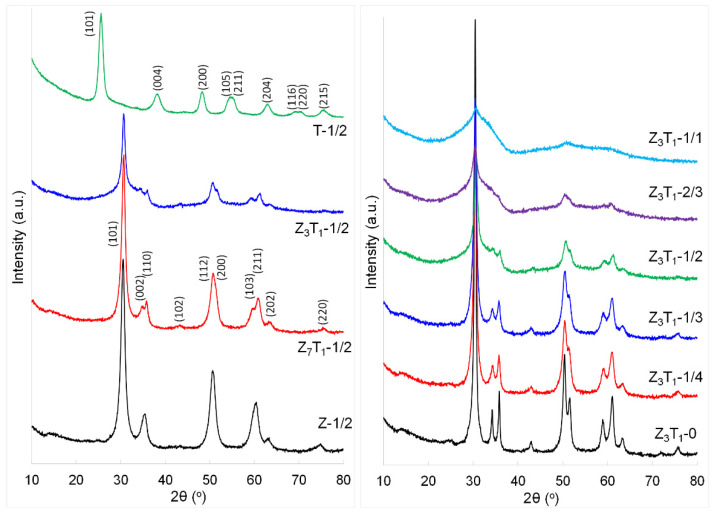
XRD patterns of the silica-leached powders.

**Figure 2 molecules-29-03278-f002:**
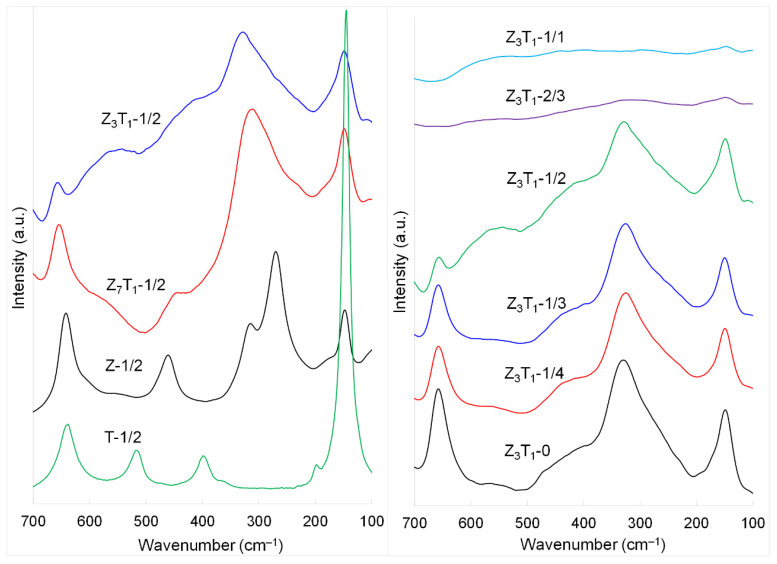
Raman spectra of the silica-leached powders.

**Figure 3 molecules-29-03278-f003:**
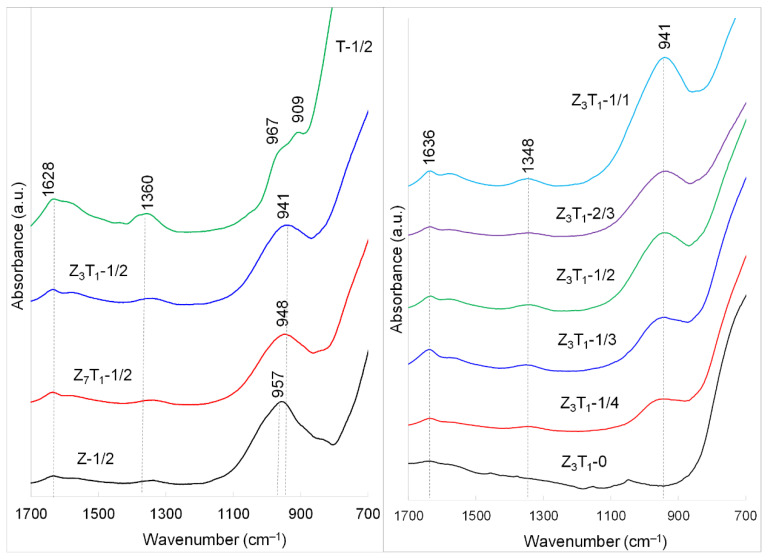
FTIR spectra of the silica-leached powders.

**Figure 4 molecules-29-03278-f004:**
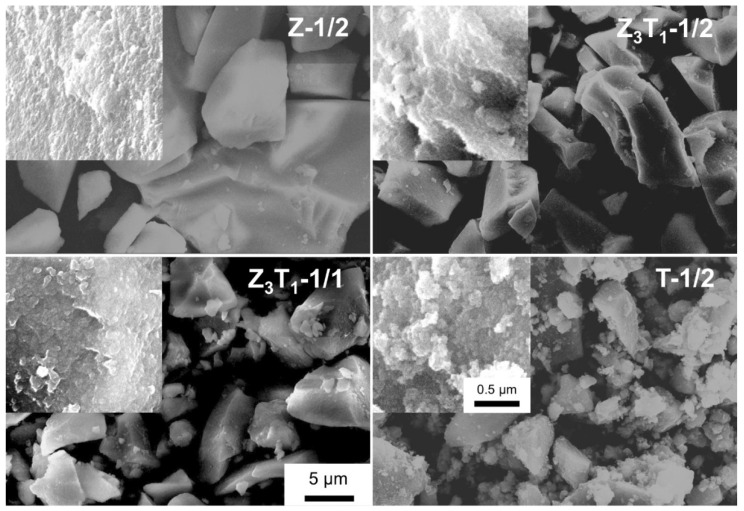
Secondary SEM images of the silica-leached powders.

**Figure 5 molecules-29-03278-f005:**
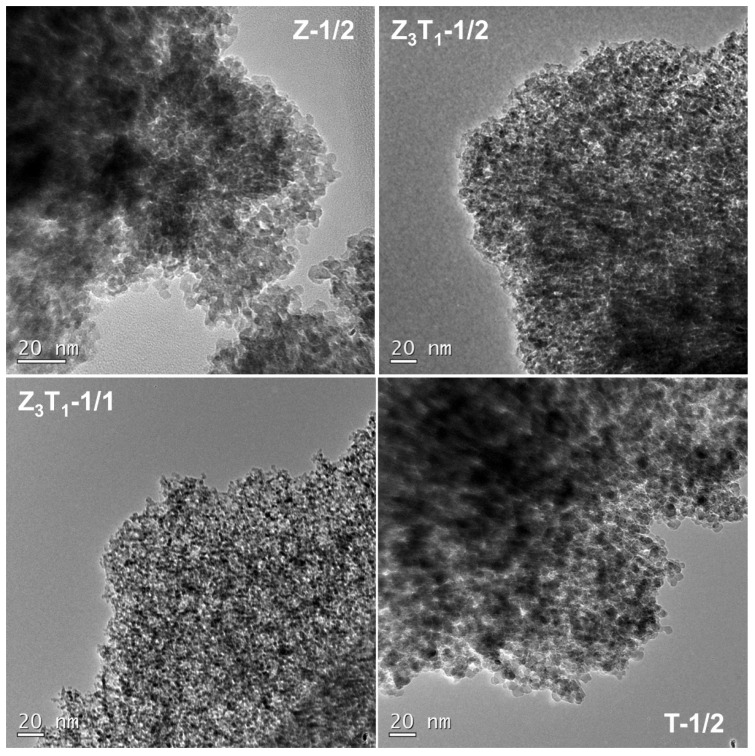
Bright field TEM images with inserted HRTEM images of the silica-leached powders.

**Figure 6 molecules-29-03278-f006:**
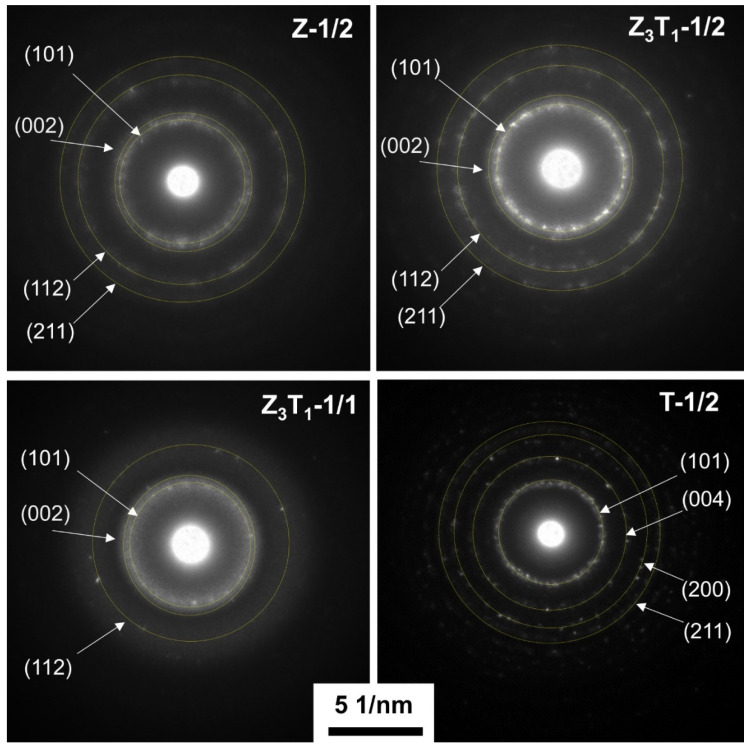
SAED patterns of the silica-leached powders.

**Figure 7 molecules-29-03278-f007:**
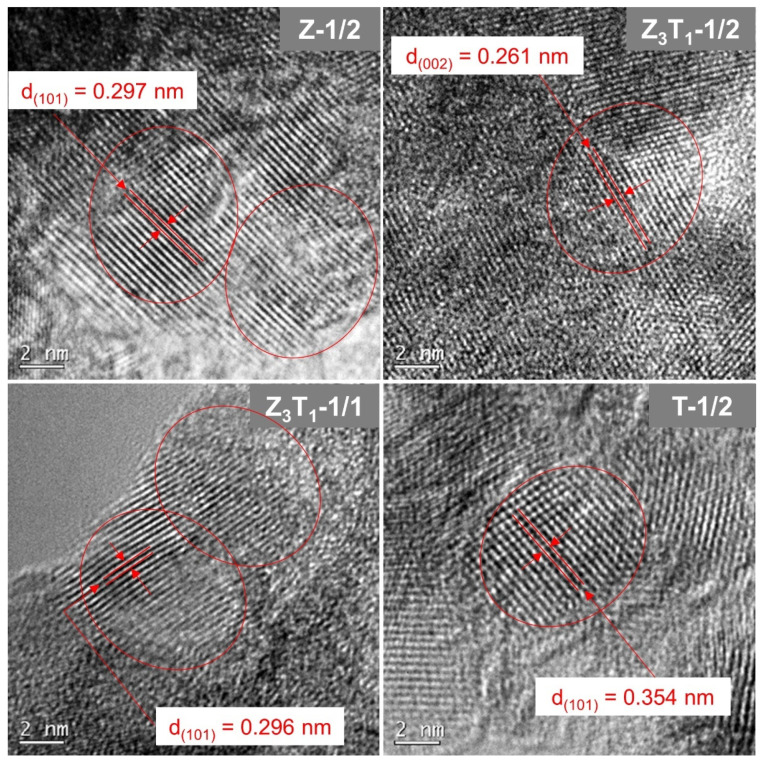
HRTEM images of the silica-leached powders.

**Figure 8 molecules-29-03278-f008:**
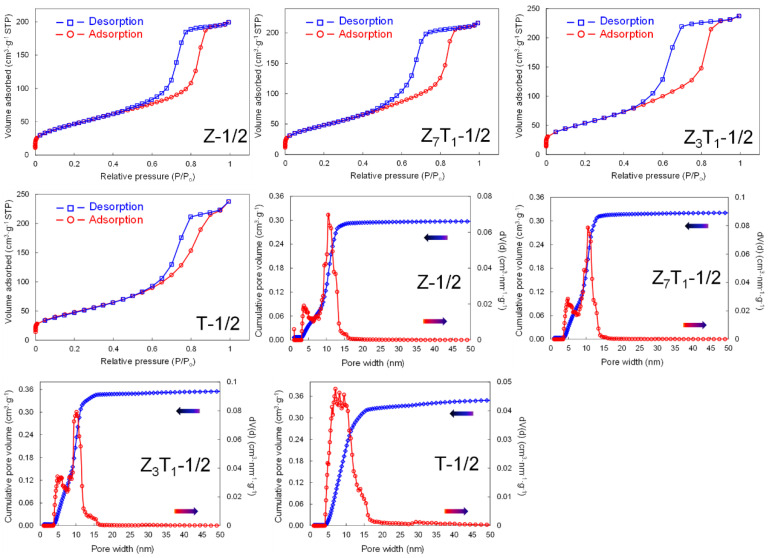
Nitrogen sorption isotherms, pore size distributions and corresponding cumulative pore volumes based on DFT analysis of the silica-leached samples, with sample descriptions shown in Table 1.

**Figure 9 molecules-29-03278-f009:**
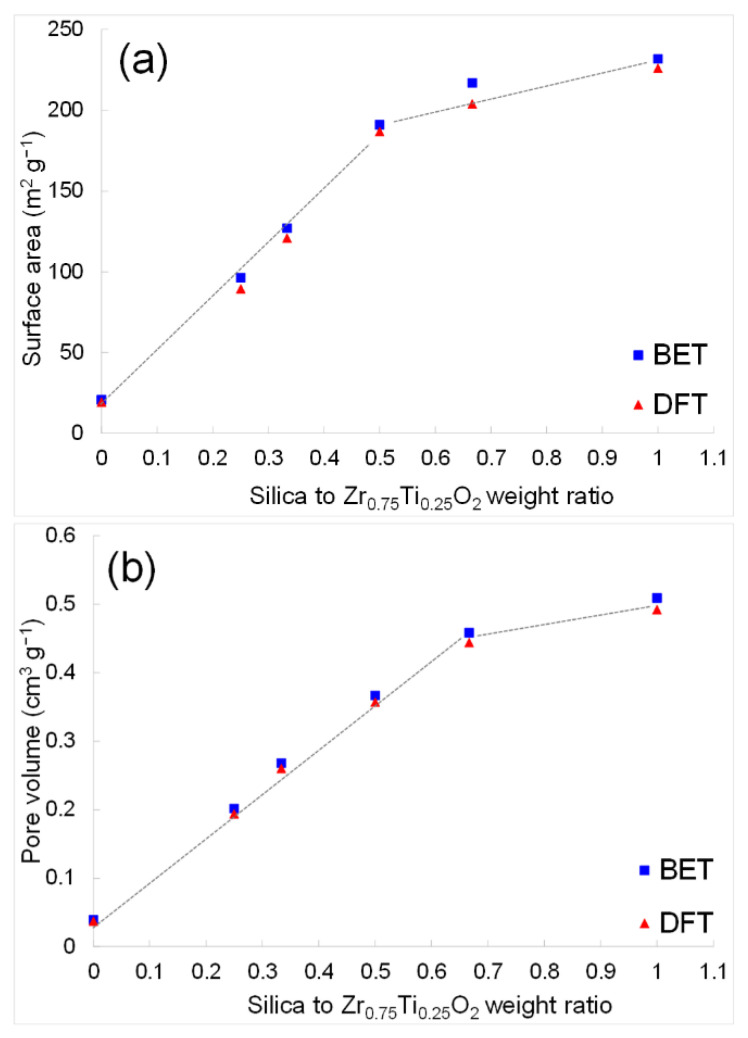
(**a**) The specific surface area and (**b**) pore volume with respect to the silica-to-Zr_0.75_Ti_0.25_O_2_ weight ratio, based on BET and DFT analyses.

**Table 1 molecules-29-03278-t001:** Sample compositions and calculated crystallite sizes using the Scherrer equation and (101) XRD peaks.

Sample	Composition	SiO_2_/ZT Weight Ratio	SiO_2_/ZT Mole Ratio	SiO_2_ Mole Fraction	*N* of ZT on Silica (nm^−2^)	Crystallite Size (nm)
Z-1/2	ZrO_2_	1:2	1.025	50.6	44	7.9
Z_7_T_1_-1/2	Zr_0.875_Ti_0.125_O_2_	1:2	0.965	49.1	47	8.9
Z_3_T_1_-1/2	Zr_0.75_Ti_0.25_O_2_	1:2	0.935	48.3	49	10
T-1/2	TiO_2_	1:2	0.665	39.9	69	7.7
Z_3_T_1_-0	Zr_0.75_Ti_0.25_O_2_	0	0	0		21
Z_3_T_1_-1/4	Zr_0.75_Ti_0.25_O_2_	1:4	0.468	31.9	97	12
Z_3_T_1_-1/3	Zr_0.75_Ti_0.25_O_2_	1:3	0.624	38.4	73	11
Z_3_T_1_-2/3	Zr_0.75_Ti_0.25_O_2_	2:3	1.247	55.5	37	4.8
Z_3_T_1_-1/1	Zr_0.75_Ti_0.25_O_2_	1:1	1.871	65.2	24	1.4 *

ZT: (Zr,Ti)O_2_ materials; *N* of ZT on silica: the number of ZT molecules per nm^2^ of silica surface; *: low confidence figure.

**Table 2 molecules-29-03278-t002:** Element analyses by EDS-SEM (mol%) (average ± stdev, N = 6) of the silica-leached (Zr,Ti)O_2_ powders using a silica-to-(Zr,Ti)O_2_ weight ratio of 1:2.

	Z-1/2	Z_3_T_1_-1/2	Z_3_T_1_-1/1	T-1/2
Zr	30.6 ± 0.9	21.6 ± 0.5	18.6 ± 0.7	
Ti		7.7 ± 0.2	7.0 ± 0.3	31.2 ± 0.5
Si	2.7 ± 0.3	2.5 ± 0.3	5.1 ± 0.4	0.7 ± 0.1
O	65.1 ± 0.5	65.3 ± 0.3	64.2 ± 0.5	65.2 ± 0.3
Na	1.6 ± 0.2	2.9 ± 0.5	5.1 ± 1.0	2.9 ± 0.7

**Table 3 molecules-29-03278-t003:** Pore structure data of the silica-leached (Zr,Ti)O_2_ powders using a silica-to-(Zr,Ti)O_2_ weight ratio of 1:2.

Sample	Z-1/2	Z_7_T_1_-1/2	Z_3_T_1_-1/2	T-1/2
BET surface area (m^2^·g^−1^)	172	176	191	171
DFT surface area (m^2^·g^−1^)	171	167	187	166
BET pore volume (cm^3^·g^−1^) *	0.309	0.335	0.367	0.367
DFT pore volume (cm^3^·g^−1^)	0.300	0.324	0.357	0.354
Average pore diameter (nm) **	7.19	7.62	7.70	8.60
DFT peak pore diameter (nm)	10.5	10.5	10.1	7.03
DFT size < 2 nm (vol%)	2.20	1.00	0.85	0.33
DFT size (2–50 nm) (vol%)	97.0	98.0	98.7	98.3

* Single-point total volume of pores at P/P_0_ > 0.99. ** Average pore diameter determined by BET (4V/A).

**Table 4 molecules-29-03278-t004:** Pore structure data of the silica-leached Zr_0.75_Ti_0.25_O_2_ powders at various silica-to-Zr_0.75_Ti_0.25_O_2_ weight ratios.

Sample	Z_3_T_1_-0	Z_3_T_1_-1/4	Z_3_T_1_-1/3	Z_3_T_1_-1/2	Z_3_T_1_-2/3	Z_3_T_1_-1/1
BET surface area (m^2^·g^−1^)	20.6	96.4	127	191	217	232
DFT surface area (m^2^·g^−1^)	19.4	89.3	121	187	204	226
BET pore volume (cm^3^·g^−1^) *	0.0390	0.202	0.268	0.367	0.458	0.509
DFT pore volume (cm^3^·g^−1^)	0.0371	0.194	0.260	0.357	0.444	0.492
Average pore diameter (nm) **	7.55	8.37	8.43	7.70	8.43	8.78
DFT peak pore diameter (nm)	7.03	10.1	10.1	10.1	10.1	10.1
DFT size < 5 nm (vol%)	0	0	0	0.85	0	0.43
DFT size (5–50 nm) (vol%)	98.3	98.5	99.2	98.7	99.4	98.5

* Single-point total volume of pores at P/P_0_ > 0.99. ** Average pore diameter determined by BET (4V/A). wt.: weight; mol.: mole; vol.: volume.

## Data Availability

All data supporting the reported results are available on request from the corresponding author.

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
