# Peer review of "Synthesis of Mesoporous Tetragonal ZrO2, TiO2 and Solid Solutions and Effect of Colloidal Silica on Porosity"

_molecules, 2024, doi:10.3390/molecules29143278_

Round 1
Reviewer 1 Report
Comments and Suggestions for Authors
The manuscript "Synthesis of mesoporous tetragonal ZrO2, TiO2 and solid..." submitted by Kong and coworkers presents synthesis of porous Zr(IV) and Ti(IV) oxide solids. The use of a Ludox template to maintain porosity while developing crystallinity in the oxide solid a 600C is a worthwhile investigation. The authors present clear crystallographic and EM evidence for their materials. The FTIR spectra in the supplemental info (and in the main manuscript) are presented in an unorthodox manner. It would be better to present as %T rather than absorbance (in addition to the fact that there is no scale on the abs axis). The spectra, particularly in the supplemental information, are poor quality and do not resemble other literature spectra for these oxides. The authors should reanalyze and modify the presentation of the IR spectra.
Comments on the Quality of English Language
The writing is not particularly clear in the manuscript, but it is not due to any language issues
Author Response
Reviewer 1:
The manuscript "Synthesis of mesoporous tetragonal ZrO2, TiO2 and solid..." submitted by Kong and coworkers presents synthesis of porous Zr(IV) and Ti(IV) oxide solids. The use of a Ludox template to maintain porosity while developing crystallinity in the oxide solid a 600C is a worthwhile investigation. The authors present clear crystallographic and EM evidence for their materials.
The FTIR spectra in the supplemental info (and in the main manuscript) are presented in an unorthodox manner. It would be better to present as %T rather than absorbance (in addition to the fact that there is no scale on the abs axis). The spectra, particularly in the supplemental information, are poor quality and do not resemble other literature spectra for these oxides. The authors should reanalyze and modify the presentation of the IR spectra.
Response: It is difficult to agree with the above comments regarding IR spectra. Though transmittance (%) is more intuitive in terms of understanding how much light is passing through the sample, and we acknowledge that some researcher use %T to plot the IR spectrum, Absorbance is often preferred in analytical chemistry because it has a linear relationship with concentration, making it easier for quantitative analysis. In fact, the IR spectroscopy theory utilizes the concept that molecules tend to absorb specific frequencies of light resulting in a characteristic pattern of absorbed frequencies called an absorbance spectrum.
Furthermore, in respective Figures the IR patterns are shifted alone the Y axis for clarity, e.g., Figure 4 in a recent work in reference 39 shown below. As such neither Absorbance nor Transmittance spectra would have a scale bar on Y axis.
[Ref-39: Ellerbrock, R.; Stein, M.; Schaller, J. Comparing amorphous silica, short‑range‑ordered silicates and silicic acid species by FTIR. Sci. Reports 2022, 12, 11708].
For Reviewer’s inspection, the IR patterns with the scale bars in both Absorbance and %T are shown below, which look messy and are difficult for discussion.
Abs with unit:
T%
Finally, we would like to thank the reviewer for the suggestions.

Reviewer 2 Report
Comments and Suggestions for Authors
This work investigates the synthesis of zirconia, titania and selected Zr-Ti solid solutions via soft chemistry route using glucose as a dispersant agent and colloidal silica as a hard template. The synthesized powders possess tetragonal crystal structure and hierarchical mesoporous features.
As stated by the authors, these mesoporous powders could be used for nuclear applications such as generator adsorbents and separations, due to the radiation tolerance of the tetragonal crystal structure and the high specific surface area.
In my opinion, the quality of the manuscript is high in terms of readibility, process explanations, characterizations performed, discussion of results and perspectives, since it covers a field (radiation tolerance for nuclear application) usually not explored using such kind of materials.
Just a comment, to further increase the readibility of the paper: I would exchange paragraph 2 and 3, introducing the synthesis explanation and the description of the employed characterization instruments before the discussion of the results.
I would also increase the resolution quality of Fig. 2, Fig. 3, Fig. 8 and Fig. 9.
Author Response
Reviewer 2:
This work investigates the synthesis of zirconia, titania and selected Zr-Ti solid solutions via soft chemistry route using glucose as a dispersant agent and colloidal silica as a hard template. The synthesized powders possess tetragonal crystal structure and hierarchical mesoporous features.
As stated by the authors, these mesoporous powders could be used for nuclear applications such as generator adsorbents and separations, due to the radiation tolerance of the tetragonal crystal structure and the high specific surface area.
In my opinion, the quality of the manuscript is high in terms of readability, process explanations, characterizations performed, discussion of results and perspectives, since it covers a field (radiation tolerance for nuclear application) usually not explored using such kind of materials.
Just a comment, to further increase the readability of the paper: I would exchange paragraph 2 and 3, introducing the synthesis explanation and the description of the employed characterization instruments before the discussion of the results.
Response: We understand that the format of most journal papers requires the “Materials and methods” section prior to “Results and discussion” section.
However, “Molecules” requires:
- Research manuscripts should comprise:
- Front matter: Title, Author list, Affiliations, Abstract, Keywords.
- Research manuscript sections: Introduction, Results, Discussion, Materials and Methods, Conclusions (optional).
- Back matter: Supplementary Materials, Acknowledgments, Author Contributions, Conflicts of Interest, References
I would also increase the resolution quality of Fig. 2, Fig. 3, Fig. 8 and Fig. 9.
Response: The above figures and Fig. 1 were replotted, and the resolution has increased.
Finally, we highly appreciate the reviewer for the valuable comments and suggestions.

Reviewer 3 Report
Comments and Suggestions for Authors
In this MS authors studied the aggregated nanoparticles of titanium and zirconium oxide solid solution of tetragonal system and obtained materials with t high specific surface area. The samples were characterized by different techniques. The is possible to be published in Molecules. But few details need be clarified.
The PXRD in this work are not very broad. Therefore, the linewidth of machine should be removed. The real crystallite size will clearly larger than what presented in this MS.
Line 149-150: “The crystallite size of t-(Zr,Ti)O2 formed on colloidal silica particle surface depends on the concentration of (Zr,Ti)O2 per nm2 of silica surface”. Authors considered “(Zr,Ti)O2 formed on colloidal silica particle surface”, but did not give evidence. The heat treatment also could induce the formation of (Si,Ti,Zr)O2 and then phase separation to (Ti,Zr)O2-rich phase and SiO2-rich phase. The SiO2-rich phase was finally removed by NaOH treatment.
The discussions between Lines 155-165 are too subjective. Authors should give experimental evidence.
Authors said “In general, the silica covers ~60–70% surface area of Z3T1 powder.” (Line 412) But it is not clear how comes this comment. I suggest authors giving more details on the way of estimations collected in Table S1.
English need be improved.
For example, it could use “crystalline ternary metal oxides” to replace “ternary metal oxides possessing crystal structure (line 420)”
Line 65: I think “(Y2Ti2O7)” should be called “binary metal oxide” or “ternary compound” not “ternary metal oxide”.
Comments on the Quality of English Language
Minor editing of English language required.
Author Response
Reviewer 3:
In this MS authors studied the aggregated nanoparticles of titanium and zirconium oxide solid solution of tetragonal system and obtained materials with high specific surface area. The samples were characterized by different techniques. The is possible to be published in Molecules. But few details need be clarified.
The PXRD in this work are not very broad. Therefore, the linewidth of machine should be removed. The real crystallite size will clearly larger than what presented in this MS.
Response: We accept the criticism from the reviewer that Powder X-Ray diffraction analysis (PXRD) using Cu Ka radiation (lav = 1.54187 Å) is not an ideal technique to analyze the crystal structure and estimate the crystallite size. Synchrotron X-ray diffraction (SXRD) or XRD Neutron Diffraction should be utilized. However, powder X-ray diffraction (XRD) patterns in the 2q range of 10–80o may be acceptable for analyzing the tetragonal zirconia with main tetragonal structure reflections at (101), (002), (110), (102), (112), (200), (103), (211), (202), (220) in the range of 25–80o (2θ). Similar study is reported in reference-8 [Abel, K.L.; Weber, S.; Poppitz, D.; Titus, J.; Sheppard, T.L.; Gläser, R. Thermally stable mesoporous tetragonal zirconia through surfactant-controlled synthesis and Si-stabilization. RSC Adv. 2022, 12, 16875–16885].
Line 149-150: “The crystallite size of t-(Zr,Ti)O2 formed on colloidal silica particle surface depends on the concentration of (Zr,Ti)O2 per nm2 of silica surface”. Authors considered “(Zr,Ti)O2 formed on colloidal silica particle surface”, but did not give evidence. The heat treatment also could induce the formation of (Si,Ti,Zr)O2 and then phase separation to (Ti,Zr)O2-rich phase and SiO2-rich phase. The SiO2-rich phase was finally removed by NaOH treatment.
Response: Figure S1 in Supplementary Materials (SI) shows the XRD patterns of the calcined powders prior to leaching silica hard templates. These patterns are similar to those of the counterpart samples shown in Figure 1 (XRD patterns of the silica leached powders), suggesting silica is in amorphous state and does not incorporate inside the tetragonal crystal structure. It is probably indirectly evidenced that the presence of silica has no impact for the formation of the tetragonal (Zr,Ti)O2 materials.
There are no search results from the ICSD for (Si,Ti,Zr)O2 compositional crystal materials. In addition, the calcination temperature is only 600 °C, it is less likely that Si is incorporated into (Zr,Ti)O2 tetragonal cells.
Because the silicon atom is small and strongly prefers tetrahedral coordination, its dissolution within the ZrO2 lattice seems less probable than at the superficial location [22]. It is evidenced [22] that the zirconia textural can be stabilized by siliceous species, and this effect is due to smearing of the siliceous species over the precipitate surface. This support surface is strongly enriched by silicon; at a Si content as low as 4 wt%, the solids show properties similar to pure silica catalytic supports for both noble metal- and sulfide-catalyzed reactions [22], indicating Si-stabilized zirconia possessing substantial surface area and thus high concentration of active sites (−OH) [8].
These discussions have been mentioned in manuscript.
The discussions between Lines 155-165 are too subjective. Authors should give experimental evidence.
Response: The discussions between Lines 155-165 are: “The formation mechanisms of the t-ZrO2 or t-Z3T1 in presence of Si may be different between Si introduced as colloidal silica (SiO2) and silicon alkoxide at molecules level. When silica nanoparticles are used, the interaction between Si and Zr or Zr-Ti occurs on the surface of the silica colloids. The silanol groups (Si−O−H) on silica surface are functional active sites for any chemical reactions. The hydroxyl groups (−OH) on colloidal silica react with hydroxyl groups on zirconia (Zr−OH) followed by dehydration condensation reaction, resulting in the formation of chemical bonds between ZrO2 and silica (Zr−O−Si). Under the same weight of silica, it is expected that the smaller the colloidal particles, the higher the surface area, and as such the more reaction sites. In this case, under fixed amount of the Zr or Zr-Ti molecule, there will be more crystal seeds formed on silica surface, as a result, smaller crystallite or crystal grains will be produced”.
We appreciate reviewer’s comment and understand reviewer’s concern that we should not discuss the silica size effect without the experimental results. As a result, we re-write the sentences highlighted above as following:
“In this case, the more silica colloids are presented, the higher the surface area of the silica is, as such the more reaction sites are. Under fixed amount of the Zr or Zr-Ti molecule, less Zr or Zr-Ti reactants presented in a fixed silica surface area leads to smaller crystallite or crystal grains”.
The discussion is supported by the experimental results displayed in Table 1 and Table 4.
Authors said “In general, the silica covers ~60–70% surface area of Z3T1 powder.” (Line 412) But it is not clear how comes this comment. I suggest authors giving more details on the way of estimations collected in Table S1.
Response: The following paragraph has been added in SI for clarification.
For 1 g of the crystalline ZT, the total volume of ZT particles (VTot) can be obtained using density ρ being 5.638 g cm–3. It is assumed that the ZT crystallite is spherical and crystallite size equals to the particle diameter, so both the volume per ZT particle (Vparticle) and the surface area per ZT particle (Aparticle) can be calculated using the crystallite size in Table 1. The total number of ZT particles (NTot) equals to VTot/Vparticle. The calculated surface area of the ZT material (Acal) is Aparticle x NTot, and the experimental surface area of the ZT material (AExp) is the nitrogen adsorption analysis results by DFT modelling (Table 4). Amorphous colloidal silica (Ludox® HS-30) has surface area ~220 m2 g–1. Based on the silica to ZT weight ratio, the silica mass and then the calculated surface area of the silica (ACal-silica) are obtained. As a result, both the silica to calculated ZT surface area ratio and the silica to experimental ZT surface area ratio can be estimated.
English need be improved.
Response: We have endeavored to improve the English which has been carefully checked.
For example, it could use “crystalline ternary metal oxides” to replace “ternary metal oxides possessing crystal structure (line 420)”
Response: “ternary metal oxides possessing crystal structure” is replaced by “crystalline ternary metal oxides”, according to reviewer’s advice.
Line 65: I think “(Y2Ti2O7)” should be called “binary metal oxide” or “ternary compound” not “ternary metal oxide”.
Response: We believe that the ternary metal oxides are designated as AxByOz.
https://en.wikipedia.org/wiki/Ternary_compound
A ternary compound of type A2BX4 may be in the class of olivine, the spinel group, or phenakite. Examples include K2NiF4, β-K2SO4, and CaFe2O4.
https://en.wikipedia.org/wiki/Oxide
This applies to binary oxides, that is, compounds containing only oxide and another element.
At last, we sincerely appreciate the reviewer for the valuable comments and advice to improve the manuscript.

Reviewer 4 Report
Comments and Suggestions for Authors
The manuscript entitiled "Further increasing silica equaling to Z3T1 in weight leads to a very low crystallized" described the preparation of mesoporous metal oxide material and evaluated the influence of the preparation parameters. The topic is interesting, however, there are many concerns should be addressed.
Comment 1: Please specify the novelty more clearly in the introduction part, since mesoporous metal oxided obtained from hard template has been reported extensively.
Comment 2: Why the hard template influence the crystallinity regarding "further increasing silica equaling to Z3T1 in weight leads to a very low crystallized"?
Comment 3: It is suggested to describel the name of the samples in the Methods part more clealy.
Comment 4: Is there any SiO2 remaining in the etched sample? What is the Si content for the samples? ICP characterization is suggested.
Comment 5: The authro stated that Si-OH is important for the adsorption application, how about the function of Ti-OH, Zr-OH? Please distinguish between these groups.
Comment 6: Please measure the pore size of the samples in the TEM characterizations.
Comment 7: The more recent literatures regarding to the mesoporous metal oxides are suggested to be cited: Chemical Engineering Journal, 494 (2024) 153028, Journal of Material Chemistry A, 2024, DOI: 10.1039/D4TA01645A
Comments on the Quality of English Language
The english quality is ok.
Author Response
Reviewer 4:
The manuscript entitled "Further increasing silica equaling to Z3T1 in weight leads to a very low crystallized" described the preparation of mesoporous metal oxide material and evaluated the influence of the preparation parameters. The topic is interesting, however, there are many concerns should be addressed.
Comment 1: Please specify the novelty more clearly in the introduction part, since mesoporous metal oxides obtained from hard template has been reported extensively.
Response: We agree with the reviewer that the synthesis of the mesoporous metal oxides using hard template has been reported previously. However, to our best knowledge, this work pursues the initial study to synthesize the hierarchical mesoporous zirconia, titania, and Zr-Ti solid solutions possessing a tetragonal crystal structure, by employing a ‘soft’ dispersant agent and a hard silica template. This tetragonal crystal phase is required when the materials are used as adsorbents for radioisotope generators and separation applications due to its high radiation resistance feature. In addition, relatively satisfactory porosity characteristics including specific surface area up to ~200 m2 g–1, pore volume up to ~0.5 cm3 g–1, and pore size ~5–20 nm are achieved.
All these features have been elaborated in “Introduction” and “Conclusions” sections.
Comment 2: Why the hard template influence the crystallinity regarding "further increasing silica equaling to Z3T1 in weight leads to a very low crystallized"?
Response: We have thoroughly discussed the influence of the silica hard template on the crystallinity and crystallite size of the (Zr,Ti)O2 materials in manuscript as shown below.
Previous studies [30–33] show that the introduction of silicon is able to stabilize t-ZrO2. The stabilization is attributed to either the constraining effect of the amorphous silica on ZrO2 particles at high SiO2 content (> 30% mol) [30,31] or the decreased kinetics of ZrO2 grain growth induced by silica as low as 2 mol% SiO2 [32,33]. High temperature calcination normally leads to the formation of m-ZrO2 [27]. When small amount of silica is introduced, it does not crystallize but forms silica-rich glassy phase concentrates in the triple junctions of the m-ZrO2 crystallites [34]. This amorphous silica-rich phase contains many substitution defects and might have considerable acidity [35]. So the introduction of silica will increase the phase transformation temperature from tetragonal to monoclinic.
Because the silicon atom is small and strongly prefers tetrahedral coordination, its dissolution within the ZrO2 lattice seems less probable than at the superficial location [22]. It is evidenced [22] that the zirconia textural can be stabilized by siliceous species, and this effect is due to smearing of the siliceous species over the precipitate surface. This support surface is strongly enriched by silicon; at a Si content as low as 4 wt%, the solids show properties similar to pure silica catalytic supports for both noble metal- and sulfide-catalyzed reactions [22], indicating Si-stabilized zirconia possessing substantial surface area and thus high concentration of active sites (−OH) [8]. This observation is very important for further reaction with other species via either hydrogen bonding or electrostatic force.
The formation mechanisms of the t-ZrO2 or t-Z3T1 in presence of Si may be different between Si introduced as colloidal silica (SiO2) and silicon alkoxide at molecules level. When silica nanoparticles are used, the interaction between Si and Zr or Zr-Ti occurs on the surface of the silica colloids. The silanol groups (Si−O−H) on silica surface are functional active sites for any chemical reactions. The hydroxyl groups (−OH) on colloidal silica react with hydroxyl groups on zirconia (Zr−OH) followed by dehydration condensation reaction, resulting in the formation of chemical bonds between ZrO2 and silica (Zr−O−Si). Under the same weight of silica, it is expected that the smaller the colloidal particles, the higher the surface area, and as such the more reaction sites. In this case, under fixed amount of the Zr or Zr-Ti molecule, there will be more crystal seeds formed on silica surface, as a result, smaller crystallite or crystal grains will be produced.
Comment 3: It is suggested to describel the name of the samples in the Methods part more clealy.
Response: The sample names, the corresponding material compositions, and silica to Zr,Ti)O2 weight ratios are demonstrated in Table 1. Each sample name indicates the Zr to Ti mole ratio and silica to (Zr,Ti)O2 weight ratio, which leads us to say that sample name is properly described.
|
Sample |
Composition |
SiO2/ZT weight ratio |
|
Z-1/2 |
ZrO2 |
1:2 |
|
Z7T1-1/2 |
Zr0.875Ti0.125O2 |
1:2 |
|
Z3T1-1/2 |
Zr0.75Ti0.25O2 |
1:2 |
|
T-1/2 |
TiO2 |
1:2 |
|
Z3T1-0 |
Zr0.75Ti0.25O2 |
0 |
|
Z3T1-1/4 |
Zr0.75Ti0.25O2 |
1:4 |
|
Z3T1-1/3 |
Zr0.75Ti0.25O2 |
1:3 |
|
Z3T1-2/3 |
Zr0.75Ti0.25O2 |
2:3 |
|
Z3T1-1/1 |
Zr0.75Ti0.25O2 |
1:1 |
Comment 4: Is there any SiO2 remaining in the etched sample? What is the Si content for the samples? ICP characterization is suggested.
Response: The residue SiO2 remained in etched samples is analyzed by SEM-EDS which is expressed in Table 2. Due to the crystallization of the ZrO2, TiO2, or Zr-Ti solid solutions, it would be extremely difficult to completely dissolve these crystalline metal oxides in an acidic solution, prior to ICP-MS analyzing.
Comment 5: The authro stated that Si-OH is important for the adsorption application, how about the function of Ti-OH, Zr-OH? Please distinguish between these groups.
Response: As explained in experimental section, the dried gel is calcined at 600 °C. During this procedure, the materials (except silica) are crystallized, and the dihydroxylation occurs through which the hydroxyl groups (–OH) on ceramics are released by forming a water molecule. As a result, the –OH groups attached on ZrO2, TiO2, or Zr-Ti solid solution surface are very limited, if there are any.
In the following leaching process, the majority of amorphous SiO2 is etched leaving small amount of the siliceous species smearing on ZrO2, TiO2, or Zr-Ti solid solutions, which is discussed in literature [refs: 8 and 22] and Raman results in this work.
Comment 6: Please measure the pore size of the samples in the TEM characterizations.
Response: To our understanding, TEM imaging is usually used to observe the micro-morphology of a material, and TEM-SAED and HR-TEM are used to characterize the crystal micro-structure. Certainly, pore structure can be observed by TEM in a small domain, however, we would say it is not an ideal technique to analyze the porosity and any discussion about the pores in a quantitative manner derived from TEM observation could be questionable.
Comment 7: The more recent literatures regarding to the mesoporous metal oxides are suggested to be cited: Chemical Engineering Journal, 494 (2024) 153028, Journal of Material Chemistry A, 2024, DOI: 10.1039/D4TA01645A.
Response: Literature “Meng, W.; Song, X.; Bao, L.; Chen, B.; Ma, Z.; Zhou, J.; Jiang, Q.; Wang, F.; Liu, X.; Shi, C.; Li, X.; Zhang H. Synergistic doping and de-doping of Co3O4 catalyst for effortless formaldehyde oxidation. Chem. Eng. J. 2024, 494, 153028” is cited in revised manuscript.
Finally, we sincerely appreciate the reviewer for constructive comments to improve the manuscript.

Round 2
Reviewer 4 Report
Comments and Suggestions for Authors
The author has well addressed my comments, the manuscript is suggested to be accepted.